# Word Embeddings as Statistical Estimators

## Abstract

Word embeddings are a fundamental tool in natural language processing. Currently, word embedding methods are evaluated on the basis of empirical performance on benchmark data sets, and there is a lack of rigorous understanding of their theoretical properties. This paper studies word embeddings from the theoretical perspective of statistical inference, which is essential for formal inference and uncertainty quantification. We propose a copula-based statistical model for text data and show that under this model, the now-classical Word2Vec method can be interpreted as a statistical estimation method for estimating the theoretical pointwise mutual information (PMI). Next, by building on the work of Levy & Goldberg (2014), we develop a missing value-based estimator as a statistically tractable and interpretable alternative to the Word2Vec approach. The estimation error of this estimator is comparable to Word2Vec and improves upon the truncation-based method proposed by Levy & Goldberg (2014). The proposed estimator also compares favorably with Word2Vec in a benchmark sentiment analysis task on the IMDb Movie Reviews data set.

## 1 Introduction

In natural language processing (NLP) the notion and construction of an *embedding* (i.e., a mapping of a linguistic object such as a word, sentence, or entire document to a vector in Euclidean space) is the essential link for making precise the features of language that we hope to learn or understand with statistical or machine learning tools. *Word embeddings*, arguably introduced by Deerwester et al. (1990), have grown greatly in popularity since their utility in downstream tasks were demonstrated by Collobert & Weston (2008). Specifically, the Word2Vec algorithm (Mikolov et al., 2013) greatly influenced the use of word embeddings by providing a fast and effective unsupervised approach to constructing word embeddings. This was quickly followed by the GloVe algorithm (Pennington et al., 2014), which had performance comparable to Word2Vec. More modern deep learning word embedding algorithms, such as ELMo (Peters et al., 2018), BERT (Devlin et al., 2019), and GPT-3 (Brown et al., 2020), have further pushed the performance of word embeddings to state-of-the-art levels on a variety of downstream tasks such as masked language modelling, part-of-speech tagging, analogy completion, and text generation. Though these powerful techniques have already demonstrated far-reaching societal impacts, it remains important to understand the nature of the embeddings that underlie their success.

All of the aforementioned embedding techniques employ some strategy to generate word embeddings from a corpus based on some intuited feature of natural language, theoretically giving the word embeddings relational meaning. Some examples of such strategies include the order of words in a sentence, how often pairs of words appear near one another, and the number of times certain words occur in various documents. However, these algorithms are *not* ultimately evaluated on how well they represent the intuited features of natural language that they are designed to learn; NLP algorithms are typically only judged by their success on downstream tasks. In fact, due to a lack of any precise mathematical formulation of theoretical features that govern the meaning of natural languages, it is not clear that embeddings have any significance beyond auxiliary data features that are useful for training NLP models on downstream tasks (such as those tested by the GLUE benchmark of Wang et al., 2019, evaluating on sentiment classification, semantic equivalence evaluation, text similarity evaluation, and recognition of textual entailment, among others).

We argue that to begin to understand the natural language features that embeddings represent, we must first mathematically formalize the precise features of natural language that we conjecture to make inference on. Second, we must be able to generate synthetic natural language data that exhibit the precise features that have been formulated (i.e., there must exist a generative model). And third, we must be able to demonstrate theoretical and/or empirical statistical consistency of estimation procedures designed to learn the features. Without any understanding of how natural language can be generated, attempting to gain insight into learned features—much less gain statistical guarantees on estimation properties for "true" underlying linguistic features—is hopeless.

The contributions of our paper are as follows:

- We consider a theoretical formulation of the skip-gram algorithm used for training unsupervised word embeddings, as in Word2Vec. It is established in Levy & Goldberg (2014) that the skip-gram algorithm minimizes its loss function at the pointwise mutual information (PMI), but the statistical properties of the estimated embedding features are not investigated. We investigate the statistical properties via simulation studies by proposing a copula-based statistical model for natural language data that truly has a given PMI matrix as a feature of the generative model.

- We provide a solution to the problem of constructing a singular value decomposition (SVD) of the PMI matrix from real natural language text data, which exhibits many infinite-valued components. Levy & Goldberg (2014) propose ad hoc truncation rules, but we adapt recent developments in missing values SVD (MVSVD) algorithms that rely on expectation-maximixation based imputation and estimation for matrices with missing components. Moreover, the right and left singular vectors resulting from the SVD of a PMI matrix can be used as embedding vectors with comparable meaning to the embedding vectors trained from the skip-gram algorithm and with similar performance in training on downstream tasks, but with improved interpretability. This is implicit in the fact that matrix decompositions such as SVD are relatively well-understood mathematically, unlike unsupervised learning algorithms.

- The copula-based statistical model that we propose is motivated by linguistic literature on theoretical distributional features of data from natural language, namely that ordered word frequencies in text corpora are often Zipffian distributed. We illustrate that it is possible to build generative models for natural language data that can be used to study theoretical features not limited to the PMI matrix.

- While the construction of embedding vectors in NLP contexts is almost exclusively driven by training models for optimized performance on downstream tasks, our work forces the question of whether embedding vectors have inherent meaningfulness for representing features of natural language.

The organization of our paper is as follows. An overview of existing language modelling approaches is given in Section 2. This is followed by a brief discussion of the theoretical properties of the Word2Vec algorithm and our proposed model for natural language in Section 3. In Section 4, we discuss various estimators that can act as alternatives to skip-gram embeddings, examining previously used proposals as well as our new algorithm. Section 5 analyses Word2Vec and its alternatives in terms of their abilities as statistical estimators. Finally, Section 6 analyzes the performance of these algorithms on a standard sentiment analysis task to illustrate that similar behavior as statistical estimators yields similar performance in downstream tasks.

## 2    Existing Approaches

A great deal of work has already been done in the area of natural language generation and language modelling. A standard model for text generation is the $n$-gram model, in which words are generated using an $n$th order Markov chain. These $n$-gram models are popular due to their simple nature allowing for ease of statistical analysis; however, a naive $n$-gram model, creating transition probabilities by using observed frequencies from a training corpus, often fails to capture desirable linguistic features. To remedy this limitation, a variety of methods have been proposed. A common approach to creating more sophisticated $n$-gram

models is smoothing—adjusting maximum likelihood estimates for probabilities by making the distribution of word occurrence probabilities more uniform (Chen & Goodman, 1999). Examples include Good-Turing estimation (Good, 1953), Jelinek-Mercer smoothing (Jelinek & Mercer, 1980; Brown et al., 1992a), and Bayesian smoothing (Nádas, 1984; MacKay & Bauman Peto, 1995), among others. However, most attempts at smoothing still fail to take into account how similar words occur in similar contexts; class-based $n$-gram models (Brown et al., 1992b) address this by separating words into distinct classes based on frequency of co-occurrence with other words. The primary limitation of such an approach is in the difficulties encountered when words can have disparate meanings and appear in wildly different contexts.

Other approaches more sophisticated than the $n$-gram model also exist. Neural network approaches to language modelling, such as those inspired by Bengio et al. (2003), address many of these issues and perform admirably in generating reasonably likely text (as measured in Bengio et al. (2003) by the perplexity metric). However, jumping to neural networks again decreases interpretability of the model and makes proving theoretical results difficult. Another approach is found in the log-linear models of Mnih & Hinton (2008) and Arora et al. (2016). These models do provide sophisticated statistical models for language that yield interesting theoretical results, but are tailored to capturing specific linguistic features and do not generalize easily to multiple features.

## 3 Statistical Framework

### 3.1 Word2Vec and Pointwise Mutual Information

Given a training corpus, Levy & Goldberg (2014) proved that the Word2Vec algorithm, using a skip-gram algorithm with one negative sample, implicitly factors the empirical PMI matrix, given by

$$\text{PMI}(w, c) = \log \frac{\Pr(w, c)}{\Pr(w)\Pr(c)}$$

where $w$ and $c$ together form a word-context pair, $\Pr(w, c)$ is the probability of drawing $(w, c)$ as a word-context pair from the corpus, and $\Pr(w)$ and $\Pr(c)$ are the probabilities of drawing $w$ and $c$ respectively from the corpus. That is, Word2Vec generates a matrix $W$ of word embeddings and a matrix $C$ of context embeddings, and its objective function attains a global maximum when $W$ and $C$ are such that $WC^\top = \text{PMI}$. More generally, Word2Vec with $k$ negative samples factors the empirical Shifted PMI (SPMI) matrix

$$\text{SPMI} = \text{PMI} - \log(k) \cdot \boldsymbol{J}$$

where $\boldsymbol{J}$ is the all-ones matrix. Note that neither the empirical PMI nor SPMI matrix is guaranteed to have only finite entries. In a finite corpus, most words do not co-occur with each other, leading to $\Pr(w, c) = 0$ for any such non-co-occuring pair and hence $\log \Pr(w, c) = -\infty$. The presence of many $-\infty$ entries in the empirical PMI matrix limits the mathematical and statistical techniques that can be used to interpret the matrix directly, as most linear algebra techniques require the entries of a matrix to come from a ring or field, but $\mathbb{R} \cup \{\pm\infty\}$ fails to even form a semigroup. Thus, a mathematical analysis of the PMI matrix and Word2Vec will require a deeper understanding of these infinite entries.

### 3.2 Two Settings for Natural Language Modelling

We identify two mutually exclusive possibilities for the nature of a language that produces infinite entries for an empirical PMI matrix: a *sparse* setting and a *dense* setting. In both settings, we interpret the observed corpus as a *sample* which is obtained from a hypothetical, infinite *population* of text data. This population-sample framework is a fundamental building block of statistics. In any statistical inference problem, the goal is to learn something about the population, e.g., the population mean. However, it is impossible to actually observe the entire population, so we use the statistical sample as a representative and finite subset of the population on which calculations can be carried out in a feasible manner, e.g., computing the sample mean. We then use the sample metric as an estimate of the unknown population metric of interest.

We now carry this intuition to text data and more specifically, co-occurrence and PMI, under sparse and dense settings. In practice, the *sample* co-occurrence matrix, i.e., the co-occurrence matrix computed from

the observed corpus, is almost always sparse, with most of its entries being zero. This implies that the corresponding terms of the PMI matrix are $-\infty$. However, this does not necessarily mean that the *population* version of the co-occurrence matrix and the PMI matrix suffer from the same issues.

In the sparse setting, there do indeed occur words (say $w$ and $c$) that never appear in the same context in the population itself; therefore, any corpus will have an infinite entry for $(w, c)$ in its corresponding empirical PMI matrix. On the other hand, the dense setting allows for any two words to appear with each other in *some* context in the population (though this context may be very rarely seen); thus, for any two words, there is a positive probability of their co-occurrence. In other words, there is a corpus that contains a finite entry in the empirical PMI matrix for the word-context pair. Note that even under the dense setting, we are likely to observe many zero entries in the observed co-occurrence matrix, since the sample might not be large enough for observed frequencies to be non-zero even though the underlying probabilities are.

Both of these settings have an underlying notion of the "truth" of the language that Word2Vec intends to train on. However, discussing "truth" in a statistical sense is hard, if not impossible, without any understanding of how the data itself is generated. Indeed, the lack of a data-generating model for NLP data contributes to the need for the use of performance on downstream tasks for various NLP methods. To draw an analogy, suppose that a new method for regression is proposed. Rather than measuring the efficacy of this new method using downstream tasks, the designers could either directly prove results or at least test the method using simulated data $(X_i, Y_i)_{i=1}^n$ from $Y_i = f(X_i) + \varepsilon_i$ for some function $f$, covariates $X$, and random errors $\varepsilon_i$ from some distribution. With a data-generating model for natural language, we can proceed similarly with NLP methods.

Thus, choose some data-generating model for natural language. Then in the dense setting, the sparsity (in the sense of having many infinite entries) of the empirical PMI is an artifact of sampling from the model rather than a consequence of theoretical sparsity. On the other hand, in the sparse setting, the sparsity is an inherent attribute of language (i.e. the data generating model) itself.

Under a sparse model, many traditional mathematical approaches to interpreting the PMI are fruitless, since infinite entries are inherent to the data itself: There necessarily exists a pair $(w, c)$ such that $\Pr(w, c) = 0$, and thus $\log \Pr(w, c) = -\infty$ in both the population and empirical PMI matrix. Indeed, the objective function of Word2Vec itself (without normalization of word and context vectors) would require these words to have infinitely large dot products with each other, so even direct analysis of the algorithm proves difficult. It is due to these complications that the remainder of this paper primarily focuses on a dense model for language.

### 3.3 A Text Generation Framework

In light of the need for a data-generating model for natural language, we provide the following theoretical framework: Let $\mathscr{C}$ be an infinite sequence of tokens given by $(t_i)_{i=-\infty}^{\infty}$. We assume that $\mathscr{C}$ is "stationary" in the sense that any corpus (i.e. a finite substring from $\mathscr{C}$) we draw would form a representative sample—compare this to a non-stationary $\mathscr{C}$ that adjoins a fantasy textbook with a medical textbook, where which section of $\mathscr{C}$ we draw from to sample a corpus would matter. This is inspired by the statistical framework of time series analysis, where the observed data is assumed to be a finite, continuous subset of an infinite temporal sequence. Every finite sequence of tokens can be assigned some probability of occurrence based on their frequency in $\mathscr{C}$; the $n$-gram model assigns the probability of the sequence of tokens $(w_i)_{i=1}^m$ to be

$$\Pr(w_1, \ldots, w_m) = \prod_{i=1}^{m} \Pr\big(w_i \mid w_{i-(n-1)}, \ldots, w_{i-1}\big).$$

Hence, an $n$th order Markov chain can be used to generate data if the transition probabilities are specified (e.g. via a transition probability matrix or tensor); for simplicity, we restrict ourselves in this paper to a unigram (first order) model. It then still remains to answer the question of how the transition probabilities should be specified to match the distributional properties of natural language.

A natural distributional property that the model's transition probabilities should match is the marginal distribution for word frequencies (i.e. the probability of occurrence for individual words). It is well-established that words in the English language tend to follow a Zipfian distribution with Zipf parameter approximately 1

(Moreno-Sánchez et al., 2016). To illustrate this phenomenon, we display the empirical frequencies of words from the Brown Corpus (Francis & Kucera, 1979) and overlay the expected frequencies from a Zipfian distribution in Figure 1. Thus, in a fixed vocabulary of $V$ unique words, the distribution of the word probabilities

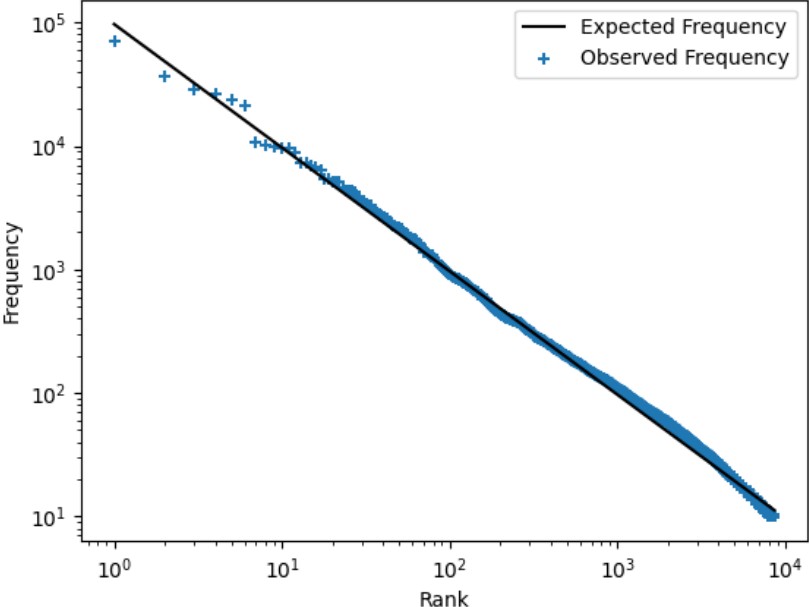

Figure 1: The empirical word frequencies from the Brown Corpus and expected word frequencies from a Zipf$(1, V)$ distribution, where $V \approx 10^4$. Infrequent words (with $< 10$ occurrences) have been omitted.

should follow a Zipf$(1, V)$ distribution (though our methodology can be extended to any distribution). This thus gives us the marginal distributions for a $V \times V$ word co-occurrence matrix; however, this does not give all the information necessary to construct the co-occurrence matrix in its entirety.

The problem of constructing a bivariate distribution when only the marginals are known is solved using *copula* distributions. Sklar's Theorem (Sklar, 1959) states that every multivariate cumulative distribution function $f_{\boldsymbol{X}}(x_1, \ldots, x_d)$ can be expressed in terms of its marginal cdfs $F_i(x_i)$ and a copula function $C$:

$$f_{\boldsymbol{X}}(x_1, \ldots, x_d) = C(F_1(x_1), \ldots, F_d(x_d)).$$

A function $C : [0, 1]^d \to [0, 1]$ is a copula function if it is a joint cumulative distribution function of a $d$-dimensional random vector on the unit cube $[0, 1]^d$ with uniform marginals.

Though Sklar's Theorem only holds for continuous distributions, many discrete multivariate distributions can still be closely approximated using copula functions. In particular, given Zipfian marginals with a Zipf parameter near 1, a Gaussian copula with correlation matrix $R$,

$$C_R(\boldsymbol{u}) = \Phi_R(\Phi^{-1}(u_1), \Phi^{-1}(u_2)),$$

where $\Phi$ denotes a normal cumulative distribution function, does well empirically in generating a bivariate Zipfian distribution with the desired marginal distributions. This thus allows us to construct a dense co-occurrence matrix, and hence (by normalizing rows to sum to unity) a transition probability matrix to generate a corpus.

Note that though this data-generating model is limited as-is, it easily extends to other, more complex situations. The current model simply generates words via a first-order Markov chain; a straightforward extension would be to use a higher order chain. This of course comes at a cost of having to work with tensors rather than matrices and greatly increases the computational cost of computing copulas. Another extension would be to use more sophisticated copulas than the Gaussian copula, allowing for more complex

structures in a language to be replicated in the model; see Ibragimov & Prokhorov (2017) for an in-depth treatment of appropriate copulas for heavy-tailed distributions such as the Zipffian distribution. Different copulas also allow for a choice between the dense and sparse settings referenced in Section 3.2.

## 4 Embedding Algorithms to Estimate the SPMI Matrix

In our theoretical framework introduced in Section 3.3, a corpus is an observed finite sample from a population of infinite length. In light of this, the empirical SPMI matrix from a corpus is simply a point estimate for the SPMI matrix associated to the population. The only reason Word2Vec factors an empirical SPMI matrix rather than the population SPMI matrix is due to the limitation of having to train on a finite corpus. Though impossible in practice, Word2Vec would ideally factor the population SPMI matrix, as this would allow for the most generally applicable word embeddings. Thus, in order to theoretically understand Word2Vec, we need methods that are easier to analyze mathematically but still approximate Word2Vec in its ability to implicitly factor the population SPMI matrix as well as in its performance on downstream tasks.

### 4.1 Truncating the SPMI

As stated in section 3.1, the objective function of Word2Vec with $k$ negative samples aims to find matrices $W$ and $C$ such that $WC^\top =$ SPMI. Hence, one can bypass the need to perform the Word2Vec algorithm by directly factorizing the empirical SPMI matrix; the simplest way to do so would be to use singular value decomposition (SVD) on the matrix

$$\text{SPMI} = U\Sigma V^\top$$

and use $U\Sigma^{1/2}$ and $V\Sigma^{1/2}$ as our word and context embeddings respectively. However, the infinite entries of the empirical PMI matrix prevent SVD from being used directly.

To circumvent the problem of the infinite entries of the empirical SPMI matrix, Levy & Goldberg (2014) instead perform SVD on the empirical *positive* SPMI (SPPMI) matrix

$$\text{SPPMI}(w, c) = \max(\text{SPMI}(w, c), 0)$$

which truncates all negative entries to 0. Though the SPPMI metric is shown to perform well empirically on semantic similarity tasks, it has the significant downside of "throwing away" a large amount of data regarding negative associations. Indeed, Table 2 of Levy & Goldberg (2014) illustrates that as the number of negative entries increases (e.g. via an increase in negative sampling for the Word2Vec procedure), performance of SVD on the SPPMI generally decreases. Similarly, Table 1 of Levy & Goldberg (2014) demonstrates that as the number of negative entries increases, SVD of the SPPMI matrix continues to deviate further from the empirical PMI. Thus, a better approach to dealing with missing co-occurrence frequencies is necessary.

### 4.2 Missing Values SVD

In the dense setting, it makes sense to look into MVSVD algorithms in order to work around the sparsity of the empirical PMI matrix. Several such algorithms are presented in Kurucz et al. (2007). One particular algorithm, yielding an Expectation-Maximization (EM) approach to MVSVD, is shown in Algorithm 1. This algorithm essentially imputes missing values by minimizing the Frobenius norm between the imputed matrix and the given matrix with missing values on the non-missing values.

These methods typically excel in the case of data that is missing completely at random; this is not the case for the empirical PMI matrix, where the smallest values are the entries most likely to be missing. Indeed, this limitation is a special case of the primary downside of the EM-MVSVD algorithm: It only aims to minimize the error on the known, non-missing values, so no information regarding the missing values is incorporated. As a result, if the matrix entries come from a known distribution, the EM-MVSVD algorithm may yield a factorization that is incompatible with the generating distribution on the missing values. This thus motivates the need to incorporate distributional information into the EM-MVSVD algorithm in certain problems.

---

**Algorithm 1** EM-MVSVD Algorithm from Kurucz et al. (2007)

---

**Require:** $W$, a matrix with missing values
**Require:** $d$, the number of singular values to keep
    $R \leftarrow \{(i,j) \,|\, W_{ij} \text{ is missing}\}$
    **for** $(i,j) \in R$ **do**
        $W_{ij} \leftarrow$ initial guess
    **end for**
    $U, \Sigma, V^\top \leftarrow \text{SVD}(W, d)$
    $\widehat{W}^{(0)} \leftarrow U\Sigma V^\top$
    **for** $(i,j) \in R$ **do**
        $W_{ij} \leftarrow \widehat{W}^{(0)}_{ij}$
    **end for**
    **for** $t = 1, 2, 3, \ldots$ **do**
        **if** converged **then**
            **return** $U, \Sigma, V^\top$
        **end if**
        $U, \Sigma, V^\top \leftarrow \text{SVD}(W, d)$
        $\widehat{W} \leftarrow U\Sigma V^\top$
        $\lambda \leftarrow \arg\min \sum\limits_{(i,j)\notin R} \left[ W_{ij} - (\lambda \cdot \widehat{W}_{ij} + (1-\lambda) \cdot \widehat{W}^{(t-1)}_{ij}) \right]^2$
        $\widehat{W}^{(t)} \leftarrow \lambda \cdot \widehat{W} + (1-\lambda) \cdot \widehat{W}^{(t-1)}$
        **for** $(i,j) \in R$ **do**
            $W_{ij} \leftarrow W^{(t)}_{ij}$
        **end for**
    **end for**

---

**Algorithm 2** DD-MVSVD Algorithm

---

**Require:** $W$, a matrix with missing values
**Require:** $\widetilde{W}$, a matrix approximating the "true" matrix from a distribution
**Require:** $d$, the number of singular values to keep
    $R \leftarrow \{(i,j) \,|\, W_{ij} \text{ is missing}\}$
    **for** $(i,j) \in R$ **do**
        $W_{ij} \leftarrow$ initial guess
    **end for**
    $U, \Sigma, V^\top \leftarrow \text{SVD}(W, d)$
    $\widehat{W}^{(0)} \leftarrow U\Sigma V^\top$
    **for** $(i,j) \in R$ **do**
        $W_{ij} \leftarrow \widehat{W}^{(0)}_{ij}$
    **end for**
    **for** $t = 1, 2, 3, \ldots$ **do**
        **if** converged **then**
            **return** $U, \Sigma, V^\top$
        **end if**
        $U, \Sigma, V^\top \leftarrow \text{SVD}(W, d)$
        $\widehat{W} \leftarrow U\Sigma V^\top$
        $\lambda \leftarrow \arg\min \sum\limits_{(i,j)\notin R} \dfrac{\left[ \widetilde{W}_{ij} - (\lambda \cdot \widehat{W} + (1-\lambda) \cdot \widehat{W}^{(t-1)})_{ij} \right]^2}{\widetilde{W}_{ij}}$
        $\widehat{W}^{(t)} \leftarrow \lambda \cdot \widehat{W} + (1-\lambda) \cdot \widehat{W}^{(t-1)}$
        **for** $(i,j) \in R$ **do**
            $W_{ij} \leftarrow W^{(t)}_{ij}$
        **end for**
    **end for**

---

Thus, our proposed methodology to factorize the population SPMI matrix is thus as follows: Given a corpus of finite length with $V$ unique words, compute the empirical co-occurrence matrix; sort the rows and columns by marginal frequencies and then normalize the matrix to form a bivariate probability distribution. Now identify each word with its rank (1 through $V$, with 1 being most frequent and $V$ being least frequent); this allows us to compute the correlation terms to use in the Gaussian copula. The copula then gives us a dense estimate for the "true" co-occurrence probabilities, which can then be transformed into an estimate for the population SPMI matrix with no missing values. We then use this estimate as the input $\widetilde{W}$ to Algorithm 2, a *distribution-dependent* MVSVD algorithm. This modifies EM-MVSVD (Algorithm 1) by minimizing a Chi-Square Goodness-of-Fit statistic with the estimated population SPMI as opposed to merely trying to match the empirical SPMI on non-missing entries.

## 5 Simulation Study

To study the effectiveness of these methods in estimating the population SPMI matrix, we generated text and compared the matrices generated by Word2Vec, SVD of the SPPMI, EM-MVSVD (Algorithm 1), and DD-MVSVD (Algorithm 2) to the population SPMI matrix.

To do so, we read in words from the Brown Corpus and sampled 500 words uniformly from the set of unique words; analogously to our proposed methodology from Section 4.2, we then created a positive transition probability matrix to generate text as well as the associated dense SPMI matrix for this set of words via a Gaussian copula. We then ran Word2Vec, EM-MVSVD, DD-MVSVD, and SVD on the empirical SPPMI matrix; each algorithm was asked to create 100-dimensional word embeddings based on a shift of 10 negative samples. The factorizations produced were then multiplied back together to form an estimate of the population SPMI matrix; the root mean square errors (RMSE) of these algorithms compared to the population SPMI matrix are shown in Figure 2.

Evidently, the SVD of the empirical SPPMI matrix is an increasingly worse approximation to the population PMI as the corpus size increases, and it yields approximations far worse than any of the other algorithms. On the other hand, the two MVSVD algorithms perform essentially identically to Word2Vec in terms of RMSE from the population SPMI matrix. This thus demonstrates, without the need to check performance on downstream tasks, that MVSVD algorithms will yield word embeddings much more similar to Word2Vec (and thus perform similarly to Word2Vec) than SVD on the empirical SPPMI matrix does.

## 6 Real Data Analysis

To provide more evidence of our claim from the end of Section 5, we now compare the performance of Word2Vec, the MVSVD algorithms, and SVD on the SPPMI matrix on a sentiment analysis task. We trained a simple sentiment analysis classifier on the IMDB movie review data set (Maas et al., 2011), which is comprised of 50,000 anonymized movie reviews split evenly between bad reviews (1-4 star ratings) and good reviews (7-10 star ratings). The goal of the task is to train a model to predict how a review rated any given movie when given that review's text. We randomly split the data set into test-train partitions, trained each embedding technique using the training partition, then trained Bidirectional Long Short-Term Memory Networks (BiLSTMs) on movie sentiment analysis. This experiment was repeated twenty times. The input to each two-layer BiLSTM (with 100 units) was a sequence of embeddings corresponding to the words contained in the review. Each model's output was a binary-classification layer using a binary-cross entropy loss function and a sigmoid activation function. To keep computations feasible, we removed stopwords as well as words with fewer than 300 occurrences, reducing the corpus to 3104 distinct words. Additionally, we zero-padded or truncated all reviews to 500 words. The use of truncation avoided the problem of vanishing or exploding gradients in long BiLSTM chains without severely compromising the data, as 95% of reviews required no truncation.

Table 1 shows the performance of each model across five distinct embedding algorithms and negative sampling levels of one, two, and five. In addition to the previously discussed algorithms, we included a one-hot encoding of the input words followed by a dense layer with 100 nodes, which serves the purpose of acting

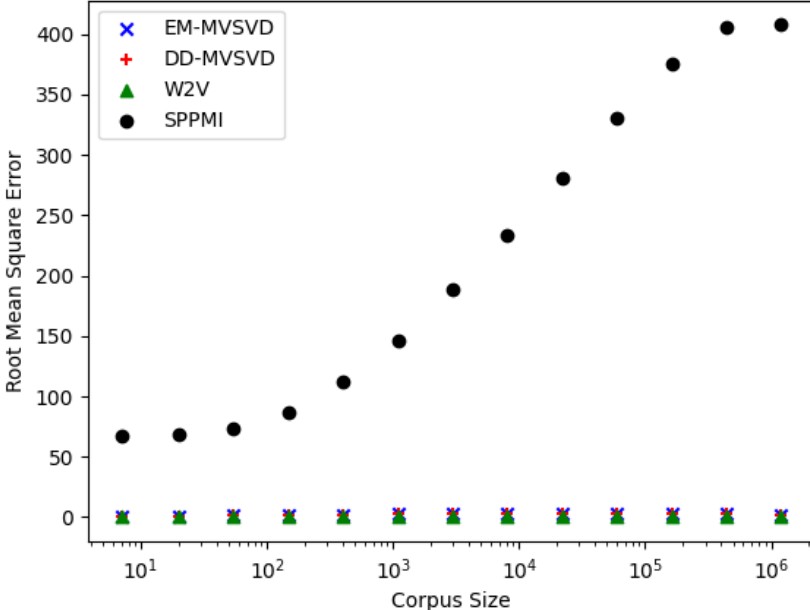

Figure 2: The RMSE of Word2Vec, the MVSVD algorithms, and SVD on the SPPMI matrix with respect to the population SPMI matrix. The RMSEs plotted are the average over 20 independent trials. Error bars are not clearly visible, as all standard errors are $< 0.5$.

| | | One-Hot | | Word2Vec | | SPPMI | | EM-MVSVD | | DD-MVSVD | |
|---|---|---|---|---|---|---|---|---|---|---|---|
| | | Acc. | S.E. | Acc. | S.E. | Acc. | S.E. | Acc. | S.E. | Acc. | S.E. |
| Negative Samples | 1 | 0.86 | 0.0047 | 0.79 | 0.0072 | 0.82 | 0.0087 | 0.82 | 0.0070 | 0.82 | 0.0146 |
| | 2 | 0.86 | 0.0065 | 0.80 | 0.0138 | 0.78 | 0.0103 | 0.81 | 0.0096 | 0.84 | 0.0034 |
| | 5 | 0.86 | 0.0028 | 0.79 | 0.0173 | 0.70 | 0.0133 | 0.80 | 0.0109 | 0.82 | 0.0114 |

Table 1: Model accuracies in positive/negative sentiment analysis on the IMDB data ste across multiple levels of negative sampling.

as a benchmark for roughly how well the "perfect" embedding overfits to the data can do in this task. All algorithms produced 100-dimensional embeddings.

As expected, the one-hot embedding performed the best, achieving 86% accuracy regardless of the amount of negative sampling. We see that the MVSVD algorithms perform comparably to Word2Vec across all levels of negative sampling, with DD-MVSVD performing the best amongst the three. We see that these algorithms still perform decently in comparison to the one-hot embeddings, achieving accuracies of roughly 80%. This is in stark contrast to the performance of SVD on the SPPMI, which matches the MVSVD algorithms at one negative sample, but quickly decreases in accuracy to 70% as the number of negative samples increases. This phenomenon occurs because the sparsity of the SPPMI matrix increases rapidly as the number of negative samples increases, so the embeddings contain less and less information to allow the BiLSTM to make a well-tuned classifier.

## 7    Conclusions

Through the use of our data-generating model for natural language, we find that the MVSVD algorithms perform similarly to Word2Vec in estimating the population PMI matrix, whereas SVD on the SPPMI matrix does not. This is reflected in the algorithms' performances on the downstream task of sentiment analysis, suggesting that an embedding algorithm's ability to estimate this population parameter strongly correlates to its performance in this particular downstream task, and perhaps to others as well. As such, the MVSVD algorithms can be seen to be quite reasonable approximations to Word2Vec that still remain tractable to mathematically analyze.

In the future, we hope to develop theory for our proposed DD-MVSVD algorithm, such as proving convergence properties of the algorithm. We also plan to investigate other more complex NLP methods and build upon our generating model to allow for more expressive modelling of language features. We also hope to be able to determine what other population parameters of natural language, if any, are correlated with performance on various downstream tasks of interest.

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
