# OpenReview forum: "Word Embeddings as Statistical Estimators"
_TMLR — Rejected by TMLR_

### Review · Reviewer_LzxB · 2022-10-26

**Summary Of Contributions:**

This paper revisited a theoretical analysis of a classical algorithm for learning word embeddings, the skip-gram algorithm, and proposed an algorithm based on a well-understood technique to learn embeddings with similar downstream performance. Building upon Levy & Goldberg (2014) findings, the authors suggested a solution to the sparsity problem when factorizing the SPMI matrix of word co-occurrences. The framework in this paper assumes a dense co-occurrence matrix where all word-context pairs occur (no matter how unlikely), and uses Missible Values SVD to solve the matrix factorization. In a simulation study, the authors showed that the skip-gram and the proposed methods successfully model language generated from a unigram model with Zipfian distribution, in contrast to the SPPMI. Revealing the statistical language feature that the skip-gram algorithm modeled. A further empirical study in a downstream task showed that the performance of the proposed method matched the skip-gram algorithm as well.

**Audience:**

Yes

**Broader Impact Concerns:**

There is no ethical concern.

**Claims And Evidence:**

No

**Requested Changes:**

1. Providing more detail on the relationship between the unigram model and copula function. Specifically, how does one obtain a transition probability from a co-occurrence matrix?
2. Providing more implications and insights from the framework. For example, you could answer the proposed question of "whether embedding vectors have inherent meaningfulness" more directly through proof or experiment.
3. Adding more experiment results.
    3.1 The simulation study does not reveal significant insights. The author could provide more analyses, such as the impact of different word frequencies, extensions to other language models, or relaxing the dense PMI assumption.
    3.2 I understand that the downstream performance is not the focus of the paper, but well-crafted downstream tasks are often used to show what information is captured by embeddings (though usually linguistic features).
4. Providing better motivation for the analyses. I think the best way to show why this kind of work is important is to show its usefulness in real applications, rather than the standpoint that we need one.

**Strengths And Weaknesses:**

Strengths
1. The paper offers a theoretical point of view and also an empirical result.

Weaknesses
1. The authors investigate the "statistical properties" of the estimated embedding, but the implication of the result remained undiscussed. In addition, the claim of improved interpretability of the proposed method was also unclear.
2. While the authors presented a success in learning word embeddings from the generated data, it was only one configuration of language modeling (unigram and zipf). It did not thoroughly show that the proposed method matched the skip-gram algorithm.
3. The skip-gram algorithm has a lasting impact on the NLP community, but the community has moved away from it. The audience of this paper would be quite limited.

---

> ### Author Response · Authors · 2022-11-05
> **Response to Reviewer LzxB**
>
> Thank you for your thorough examination of our manuscript and the helpful feedback you've given.
>
> Re: Weaknesses
> 1. We agree that the implications of our proposed data-generating model should be discussed further. We believe that adding a section concerning the implications and context (as we outline in the overall response) will be useful in addressing this concern.
>
> 2. You are correct that the current manuscript does not manage to "thoroughly" show that the MVSVD algorithms approximate Word2Vec. Unfortunately, doing so would likely have to be an entire paper in its own right.
>
> Given any dense empirical SPMI matrix (not only those generated by the unigram model with Zipfian marginals) and a fixed dimension for word embeddings, the MVSVD algorithms we've presented necessarily have an estimation error for the SPMI matrix (in terms of the Frobenius norm) that is upper bounded by that of Word2Vec by the Eckart-Young-Mirsky Theorem. Of course, a perfectly trained Word2Vec model would have zero estimation error due to its objective function in this case, so both must match.
>
> For non-dense empirical SPMI matrices, creating what amounts to concentration inequalities for the deviation between MVSVD algorithms and Word2Vec would certainly constitute an entire separate paper. The real-data analysis section was intended to demonstrate this phenomenon of approximately matching behavior on real data in lieu of formal proof. We apologize that this application was not as convincing as we had initially hoped; thus, we believe that omitting that section in favor of a further discussion of the implications of our generative model may be a net positive to the paper.
>
> 3. You are correct in saying that much of the machine learning community has moved away from applying skip-gram. However, so long as the theoretical properties remain unknown, the segment of the community interested in knowing precisely why various techniques work and how to systematically iterate upon them will continue to hold interest in older ideas that have yet to be thoroughly exhausted--including skip-gram.
>
> Re: Requested changes
>
> 1. We agree that adding more detail here can be useful; we have updated Section 3.3 of the manuscript. A transition probability matrix can be obtained from a bivariate probability mass table by normalizing each row to sum to 1.
>
> 2. We agree that a greater focus on insights and implications of the framework is necessary; we outline our plan for this discussion in the overall response. We believe that any "inherent meaningfulness" of an embedding algorithm is intrinsically related to their statistical estimation properties--that is, which textual relationships and features they aim to estimate given some fixed model for language. Our paper proposes an embedding method that directly estimates one such feature of natural language (the population PMI). Other embedding techniques, such as BERT or ELMO, have shown their effectiveness in outside experiments, but no analysis is performed to determine which language characteristics are represented within the embedding. Because each embedding technique is substantially different from one another, it is impossible to make a statement regarding the inherent meaningfulness of all embedding vectors. However, it is likely that each individual technique that has shown success does estimate some meaningful property of language (e.g. PMI for Word2Vec, log-counts for GloVe).
>
> 3/4. We disagree here on the importance of real applications; in fact, we are considering the removal of the real-data experiment altogether (see the overall response).

---

> > ### Comment · Reviewer_LzxB · 2022-11-23
> > **Reponse**
> >
> > Thank you for addressing my concerns.
> >
> > I am not concerned about your plan to remove the sentiment analysis experiment. It would help the audiences focus on the central message of the paper because finding a real-world dataset that relied solely on PMI properties would not be easy.
> >
> > However, my concern also involved the generative model framework that had only been tested with the unigram-Zipf experiment. Adding more experiments here would strengthen the contribution of the generative model framework (as shown through Word2Vec and MVSVD). I think your plan on adding more simulation studies would be helpful.

---

### Review · Reviewer_2wfU · 2022-10-26

**Summary Of Contributions:**

This paper addresses the zero-count problem in a matrix of Shifted PMI (SPMI) (Levy and Goldberg, 2014). Specifically, in a PMI matrix,

$$
{\rm PMI}(w, c) = \log \frac{P(w, c)}{P(w), P(c)} ,
$$

$\log P(w, c) = -\infty$ when $P(w, c) = 0$. These elements are actually dominant in the matrix because words $w$ and contexts $c$ rarely co-occur. We usually truncate a PMI matrix with positive elements,

$$
{\rm PPMI}(w, c) = \max({\rm PMI}(w, c), 0)
$$

However, the situation gets worse when we consider SPMI with $k$ (this corresponds to the number of negative examples in Skip-Gram with Negative Sampling),

$$
{\rm SPMI}(w, c) = {\rm PMI}(w, c) - \log k,
$$

because more elements are pushed towards zero with a larger value of $k$.

This paper proposes to use missing-value SVD algorithms to obtain the low-rank approximation of the positive SPMI matrix, approximating missing values in the matrix. This paper also proposes to use the Gaussian copula to preserve the statistical property of word occurrences. However, I could not see how this is actually realized in the study.

**Audience:**

Yes

**Broader Impact Concerns:**

None.

**Claims And Evidence:**

No

**Requested Changes:**

Major comments:

See the weaknesses.

Minor comments:

+ I don't think the explanation of the $n$-gram language model in Section 3.3 is useful because this study just considers uni-grams and Zipfician distribution.
+ $u_1$, $u_2$, and $\Phi$ are undefined in the equation of $C_R(\boldsymbol{u})$
+ Algorithm 1:
    + What is the "initial guess" in the algorithm?
    + $\| \cdot \|_{NMF}$ should be defined.
    + I'm not sure whether Algorithm 1 really corresponds to Kurucz et al. (2007). Please double-check this.

Other comments:

+ Levy and Goldberg (2014) were not the first to use positive PMI. See Bullinaria and Levy (2007).

J Bullinaria and J Levy. 2007. Extracting semantic representations from word co-occurrence statistics: A computational study. Behavior Research Methods, 39:510–526.

**Strengths And Weaknesses:**

# Strengths

+ This paper presents Expectation-Maximization based Missing-Values SVD (EM-MVSVD) and its variant, Distribution-Dependent MVSVD (DD-MVSVD) to work around the sparsity of the empirical PMI matrix of words and contexts.
+ The effectiveness of the presented algorithms is verified with the simulation study and sentiment analysis task.

# Weaknesses

+ The empirical verification is insufficient to demonstrate the superiority of the proposed method. For example, Schnabel et al. (2015) argued that there is no almighty word embeddings for all tasks; we should fine-tune word emebdddings on a target task. However, this study only reports a performance on a single task (sentiment analysis) while word embeddings are usually evaluated on multiple tasks such as word analogy, shallow parsing, and named entity recognition.
+ A comparison with GloVe may be preferrable. GloVe also has a heuristic treatment (weighting) for low co-occurrence counts, assuming that reproducing low co-occurrence counts with word embeddings is probably overkill, which somewhat contradicts with the idea suggested in this paper.
+ The detail of the proposed method is unclear. I'm not sure how the Gaussian copula is used in the proposal (although I found a short description in the end of Section 4). In addition, I could not find the detail of the BiLSTM model: how to convert encoded token emebddings into a text embedding (max-pooling?); whether the word embeddings are fine-tuned or not; etc.
+ It is hard to judge the usefulness of this study when the approach of pre-training and fine-tuning is so dominant nowadays (this is a subjective opinion).

Tobias Schnabel, Igor Labutov, David Mimno, and Thorsten Joachims. 2015. Evaluation methods for unsupervised word embeddings. In EMNLP, pages 298–307.

---

> ### Author Response · Authors · 2022-11-05
> **Response to Reviewer 2wfU**
>
> We are grateful for the time and consideration you gave to our work.
>
> Re: Weaknesses
>
> 1. We apologize for being unclear on this matter; our proposed method is not meant to be superior to Word2Vec (or any other method). Rather, our aim was to show that our method has approximately the same behavior as Word2Vec while being more tractable to analyze from a mathematical perspective. We proposed that the way to evaluate "same behavior" is via a model for natural language, which is simulated in Section 5. The purpose of the empirical section was only to demonstrate that the "same behavior" witnessed in the previous section was not an artifact of the simulated model but was also exhibited using real data.
>
> Also note that we are considering the complete removal of the real-data section in a revision; please see our overall response.
>
> 2. We agree that adding a section demonstrating our method on GloVe would be useful. This does not contradict our idea: The objective function of GloVe explicitly factorizes a log-count matrix $M$ shifted by a bias $B$ so that $W^{\top}C = M - B$. Performing a missing-values SVD on $M - B$ is well within the scope of our method, and as discussed in the overall response, we would like to add such an simulation study in a revised version of the paper.
>
> 3. We apologize that this was not clear. Given the Zipfian marginals and the correlation matrix, the output of the copula is the input $\widetilde{W}$ to Algorithm 2. We have updated the manuscript with this clarification.
>
> The BiLSTM model concatenates the forward and backward embeddings together before feeding the resulting vector into a sigmoid function for binary classification. We did not fine-tune any embeddings because each embedding was created using data from the applications domain, and we wished to compare the effectiveness of each technique directly. Furthermore, it is unclear what fine-tuning would mean for the matrix-based methods.
>
> 4. While perhaps limited in immediate application, we believe that our work is useful nonetheless; see our overall response.
>
> Re: Requested Changes
>
> 1. While the experiments only use the unigram model with Zipfian marginals and a Gaussian copula, we believe that one of the most significant contributions of our paper is the generative model itself. In order to understand statistical and machine learning models, it is invaluable to have a data-generating distribution that the working model is suited for. For example, it would be nonsensical to analyze the LASSO regression method without considering its theoretical performance on data generated by $Y = X\beta + \varepsilon$. We believe that our unigram, Zipfian marginal, Gaussian copula model is the NLP analogue of the simple linear regression model for real-valued data. A discussion of the generalizations is important in the same way that it is important to discuss generalized linear models.
>
> 2. Thank you for noting that we left $\Phi$ undefined; this is simply the Gaussian CDF. We have updated the manuscript with this change. The scalar $u_i$ is the $i$th component of the vector $\mathbf{u}$.
>
> 3. a) The initial guess can be any reasonable initialization for the iterative algorithm. Our particular implementation simply initialized with 2 less than the minimum value of the matrix--though this choice is arbitrary and up to the practitioner.
>
> b) Thank you for noting that this was not defined. NMF was intended to convey "non-missing Frobenius." We have updated Algorithm 1 with the explicit definition.
>
> c) Algorithm 1 does indeed correspond to Kurucz et al. (2007). The algorithm we display simply formalizes the discussions from sections 1.3 and 3.1 of Kurucz et al (2007).
>
> Re: Other Comments
>
> Thank you for the provided reference. While Bullinaria and Levy (2007) do use the positive PMI (PPMI) earlier, to our knowledge, the positive SPMI (SPPMI) discussed in our paper was introduced by Levy & Goldberg (2014).

---

### Review · Reviewer_isAT · 2022-10-26

**Summary Of Contributions:**

* The paper studies two methods to complete missing values in an empirical shifted pointwise mutual information (SPMI) matrix for word concurrence in a corpus and compares these to the truncation method used in Levy & Goldberg 2014

* The paper defines a generative model for text based on a unigram Markov chain using estimated vocabulary unigram distributions and a copula function, and uses text generated from this model to evaluate the performance of different word vector estimation methods (Word2Vec and SVD on different versions of empirical SPMI matrix)

* The paper shows that completing the SPMI matrix is better than the truncation method of Levy & Goldberg and similar to Word2Vec in matching PMI statistics from the generative model

* There is also evaluation on a sentiment classification task,  with similar results, but I have concerns about this application

**Audience:**

Yes

**Broader Impact Concerns:**

No concerns

**Claims And Evidence:**

No

**Requested Changes:**

* Toning down the claims regarding the method giving insights about "theoretical features that govern the meaning of natural languages"

* If possible, showing estimation error with respect to SPMI from a much larger corpus and comparing that to the copula-distribution approach.

* Showing an end task experiment following a closer to state-of-the-art formulation: e.g. word embeddings from a large unlabeled corpus followed by as many layers of LSTM or Transformer as helpful for the task.

* Adding an estimate of the computational requirements of the methods -- Word2Vec versus SVD on the positive or completed SPMI matrices.

**Strengths And Weaknesses:**

Strengths
* The motivation of this paper to have more interpretable word embeddings with better-understood statistical properties is appealing
* The paper convincingly shows that completing the SPMI matrix before performing SVD is advantageous in terms of fitting better synthetic data from the proposed generative model of text

Weaknesses
 * Since the proposed generative model of text is definitely not a good approximation of the real distribution (e.g. first order Markov chain with estimates from a finite corpus), it would be good to tone down the claims of gaining understanding with respect to real characteristics of natural language

* Would it not be feasible to evaluate various estimation methods with respect to matching PMI statistics from a much larger corpus -- e.g. estimate from 1 million tokens and see how various methods match statistics from an estimate based on 1 billion tokens? What are the advantages of using your copula-based model instead?

* In the sentiment application, word embeddings are trained on the training split of the labeled data for sentiment. This does not make sense as the advantage of the embeddings is that they can be trained don much larger unlabeled texts. I also did not understand why indicator vectors should be doing best -- they are not helping us learn about the meaning of words that are not seen in the labeled training data.

---

> ### Author Response · Authors · 2022-11-05
> **Response to Reviewer isAT**
>
> Thank you for your careful reading of our work. One point we would like to clarify from your summary is that our generative model is not limited to a unigram Markov chain--our proposal is for the use of an $n$-gram model with any marginal distribution and choice of copula that is appropriate for the scenario. We only use a unigram model with Zipfian marginals and a Gaussian copula for simplicity.
>
> Re: Weaknesses
> 1. We agree that the unigram generative model is not a good approximation of the English language's real text distribution. However, it is a clear starting point for further analysis, and it easily generalizes to act as better approximations of real language depending on the choice of order, marginal distribution, and copula. Furthermore, we believe that the data-generating model does allow us to examine features of natural language (such as the population PMI) in ways that previous classical models could not. For this reason, in the hopes of driving other researchers to take the lack of a data-generating model more seriously, we have used some strong language concerning the necessity of such a model. If, after understanding our position, you still feel that our language is strong, we are fully willing to soften instances in the minor revision (e.g. "our work *forces* the question..." -> "our work *raises* the question..." and "attempting to gain insight into learned features... is *hopeless*" -> "attempting to gain insight... is *extremely difficult*").
>
> 2. This is a good question that we agree needs to be addressed in a revision of the paper. As an analogy, suppose data points $(X_1, Y_1), \ldots, (X_m, Y_m)$ were collected as a training sample, then one evaluated various prediction methods with respect to minimizing mean-squared error (MSE). Your question is essentially asking that since one can measure MSE on 1 million samples followed by 1 billion samples, there would be no need to consider a linear model $Y = X\beta + \varepsilon$. It is true that if the only metric one cares about is performance on tasks, the underlying data-generating model doesn't matter at all. However, if one wants to truly understand what these prediction methods are doing and why they work, surely the simple linear model is the natural place to begin algorithm analysis.
>
> In the same vein, we believe that the copula-based $n$-gram model should act as an NLP analogue of the general linear model for real-valued data. The primary utility is not in evaluating whether one method is better than another, but in being able to understand the properties of these algorithms. Additionally, there is a secondary (though perhaps equally important) utility in that generative models act as a framework to systematically create new algorithms or iterate upon old ones--as opposed to needing to intuit better ones then confirm intuition via experimentation.
>
> 3. You are correct; traditionally, embeddings are created to extract information about word relationships from a large corpus of unlabeled text. Subsequent problems, such as sentiment analysis, then utilize the pre-trained embeddings to create their models. Using embeddings allows the subsequent task to work around a potentially small training set; words that do not occur in the training set can still be utilized if they appear in the embedding's lexicon. For this application, we wanted to show the impact of each embedding technique on the amount of information extracted from the source text, as applied to a small example scenario. Therefore, we decided to create embeddings directly from the IMDb training dataset and then utilize these embeddings for sentiment analysis. Any words not found within the training set were considered unknown and were zeroed out during prediction. We recognize that standard industry practice uses character embeddings or sub-word embeddings to circumvent the out-of-vocabulary problem. However, these techniques cannot be applied with Word2Vec, so we did not consider their use.
>
> In this application, the indicator vectors are performing the best because none of the chosen embedding techniques provide the meaning of words not seen in the labeled training data. All embedding techniques provide a 300-dimensional vector representation of words seen in the training data; we then evaluate how effectively these representations can perform sentiment analysis. In the sentiment analysis task, the indicator vectors can over-fit the training data, learning how best to interpret each word as a positive or negative sentiment. All other models must learn how to interpret each vector word representation as a positive or negative sentiment; these models would not be able to differentiate words with similar embeddings but different sentimentalities.

---

> > ### Author Response · Authors · 2022-11-05
> > **Response to Reviewer isAT (Continued)**
> >
> >
> > Re: Requested changes
> >
> > 1. See Weakness #1.
> >
> > 2. While we would be willing to try such an experiment, it is unclear what you mean by this. The population SPMI matrix is unavailable, as any real corpus yields only a sample SPMI matrix. Thus, one can only measure estimation error from simulated data, as done in Section 5 for corpora up to a size of $10^6$. While this simulation can be extended to even larger sample sizes, we are unsure if this would yield significantly more insight.
> >
> > 3. We appreciate the suggestion and believe it would strengthen the paper if the purpose were to demonstrate the utility of MVSVD algorithms in application. However, this is not the focus of our paper. As we mention in the overall response, we think it will be better to remove end-task experiments entirely in a later revision to prevent "real-data" experiments from acting as a distraction/a point of focus. We reiterate that the purpose of Section 6 was only to show that the claim of "same behavior" witnessed in Section 5 was not an artifact of the simulated data from the generative model but was also exhibited when using real data.
> >
> > 4. We believe that adding estimates of the computational requirements of the methods (outside of the provided supplementary material) would be counter to the paper's goal as it would indicate that such methods ought to be used in practice rather than primarily for theoretical analysis. While we would be willing to incorporate these into the manuscript in some way (e.g., promoting to an appendix rather than supplementary material), we don't think it would be useful in the main text--especially given our new plan to remove the real-data experiment from the paper (see the overall response).

---

> > > ### Comment · Reviewer_isAT · 2022-11-24
> > > **Response**
> > >
> > > Thank you for the clarifications and discussion regarding my points.
> > > About 2, I believe your simulation study measures approximation of the SPMI of your data-generating model for different methods  deriving embeddings from various-sized samples from the generative model.  I agree this experiment is useful for interpretability and insight. Nevertheless, I was suggesting an experiment on real data where we could measure the ability of different methods to approximate SPMI based on huge real-data based on SPMI statistics from smaller data. This is analogous to how prior work would measure log-likelihood from their language model on heldout data and could help justify your generative model further. If the current real data experiment could be replaced with something like this, I think this would be valuable.

---

### Author Response · Authors · 2022-11-05
**Overall Response to Reviewers**

To all the reviewers, thank you for taking the time to read our submission and provide very valuable feedback. We have updated the manuscript with the immediately addressable concerns that have been brought up, and have responded to each reviewer individually.

Across the feedback, we noticed two common trends:

1. There is an impression that the experimental results section is one of the most important parts of the paper, and the reviewers (not unjustly) found this section to be weak.

2. As it stands, the manuscript leaves an incorrect impression that the theoretical model is somehow limited to only studying Word2Vec/skip-gram and the PMI matrix. Furthermore, the importance of the generative model as a primary contribution is not evident.

In order to address the above problems during a period of "minor revisions," we plan to remove the experimental results section entirely. We believe that the current state of the machine learning community focuses on performance in downstream tasks to such a degree that its very presence acts as a distraction to the actual content of the paper.

In its place, we plan to add a section explicitly discussing the importance why a flexible generative model for language is essential to advance the systematic study of NLP, as well as further discussing the implications of our model. In essence, we believe that the generative model we propose for NLP tasks is an analogue of the generalized linear model for regression tasks--and even though all models are wrong, some are useful. If the simple linear regression model $Y = X\beta + \varepsilon$ (which no real-world data actually fits) were never created or studied, could we have developed the now-ubiquitous LASSO regression procedure? Similarly, our unigram Zipfian-marginal Gaussian-copula model is uncharacteristic of real language, but naturally yielded the new DD-MVSVD procedure that we proposed; analogously, we believe that the model will eventually allow for development of other, more powerful procedures. To quote the TMLR mission:

    "[TMLR] facilitate[s] scientific discourse on topics that are deemed less significant by contemporaries but may be important in the future"

If our goal with the generative model does not fall under this umbrella, then what does?

In order to further demonstrate the generality of our work, we also plan to demonstrate during the minor revisions period the same results using GloVe (to complement our results with Word2Vec). Since GloVe also acts as an implicit factorization of a particular population feature matrix (up to a bias term), it would be relatively straightforward to re-run the simulation of Section 5 to study how well the MVSVD algorithms approximate GloVe as well. This will yield another clear example of a "theoretical feature of natural language" (different from the PMI) that is referenced several times in the manuscript. We hope that this will assuage concerns regarding the impression that the contributions only apply to Word2Vec and the PMI matrix.

Once again, we apologize for the current framing of the manuscript that led to this confusion. We hope that the above proposed revisions will clarify the purpose and scope of our work for published version.

---

### Decision · Action_Editors · 2022-11-24

**Recommendation:** Reject

**Comment:**

The paper's contributions are of interest for the community; however the remaining reviewers' concerns cannot be addressed through a minor revision. In particular, it remains unclear whether the authors would like to keep the experimental section or remove it, and therefore the paper would need to be rewriten according to that decision in order to clearly describe the contributions and analyses of interest. See "Claims And Evidence" above for more details.

I highly encourage the authors to address these concerns and resubmit their work to TMLR.


**Audience:**

Overall, the reviewers found the contributions to be interesting. Quoting Reviewer LzxB: "the paper addressed a significant problem that had been ignored".

**Claims And Evidence:**

In the initial version of the paper, the reviewers found the experimental results to be weak, to which the authors agreed and replied they were planning to completely remove this section and focus instead of the theoretical contributions of the paper. The authors also added more results during the rebuttal period, as per the reviewers' requests.

As it stands in its second version, the reviewers think that the paper -though interesting- still needs some work before being accepted, as well as a firm decision on whether to remove the experimental results or not. If the authors decide to do so, then the theoretical analysis needs to be further emphasized; in fact one of the reviewers mentioned in the discussion that the method and its implications are still unclear. If the authors decide to do otherwise, then a few extra experiments need to be carried out (more details below).

Regarding the empirical evaluation, overall the reviewers felt that the current studies are not fully supportive of the claims. In particular, some of their concerns are:
+ The generative model framework has only been tested with the unigram-Zipf experiment (Reviewer LzxB).
+ The claim that MVSVD algorithms perform similarly to Word2Vec is not fully supported. Same comment about the generalizability of the text generation framework (Reviewer LzxB).
+ It is hard to consider the value of word embeddings without an empirical demonstration in downstream tasks (Reviewer 2wfU).
+ Add a real-data experiment (Reviewer isAT).
+ Add more simulation studies (GloVe) (Reviewer LzxB); this has been acknowledged by the authors.